# Adaptability of MODIS Daily Cloud-Free Snow Cover 500 m Dataset over China in Hutubi River Basin Based on Snowmelt Runoff Model

**Xiangyao Meng [1,2], Yongqiang Liu [1,2,*], Yan Qin [1,2], Weiping Wang [1,2], Mengxiao Zhang [1,2] and Kun Zhang [1,2]**

1   College of Geographical Science, Xinjiang University, Urumqi 830046, China;
   mengxiangyao@stu.xju.edu.cn (X.M.); qinyan0215@163.com (Y.Q.); Wangwpxju@163.com (W.W.);
   Neytiriz@163.com (M.Z.); zkun1993@163.com (K.Z.)
2   Xinjiang Key Laboratory of Oasis Ecology, Xinjiang University, Urumqi 830046, China
\*   Correspondence: lyqxju@163.com; Tel.: +86-135-7999-5658

**Abstract:** Global warming affects the hydrological characteristics of the cryosphere. In arid and semi-arid regions where precipitation is scarce, glaciers and snowmelt water assume important recharge sources for downstream rivers. Therefore, the simulation of snowmelt water runoff in mountainous areas is of great significance in hydrological research. In this paper, taking the Hutubi River Basin in the Tianshan Mountains as the study area, we used the "MODIS Daily Cloud-free Snow Cover 500 m Dataset over China" (MODIS_CGF_SCE) to carry out the Snowmelt Runoff Model (SRM) simulation and evaluated the simulation accuracy. The results showed that: (1) The SRM preferably simulated the characteristics of the average daily flow variation of the Hutubi River from May to October, from 2003–2009. The monthly total runoff was maximum in July and minimum in October. Extreme precipitation events influenced the formation of flood peaks, and the interannual variation trend of total runoff from May to October was increased. (2) The mean value of the volume difference ($D_V$) during the model validation period was 8.85%, and the coefficient of determination ($R^2$) was 0.73. In general, the SRM underestimates the runoff of the Hutubi River, and the simulation accuracy is more accurate in the normal water period than in the high-water period. (3) By analyzing MODIS_CGF_SCE from 2003 to 2009, areas above 3200 m elevation in the Hutubi River Basin were classified as permanent snow areas, and areas below 3200 m were classified as seasonal snow areas. In October, the snow area in the Hutubi River Basin gradually increased, and the increase in snow cover in the permanent snow area was greater than that in the seasonal snow area. The snowmelt period was from March to May in the seasonal snow area and from May to early July in the permanent snow area, and the minimum snow cover was 0.7%.

**Keywords:** snowmelt runoff model; MODIS snow cover dataset; Tianshan; Hutubi River

## 1. Introduction

Global snowmelt-dominated rivers are mainly distributed above 45° in the northern and southern hemispheres, and high-altitude mountainous regions are the most widely distributed areas for snowmelt runoff [1,2]. Snowpack and glaciers are natural freshwater reservoirs that provide stable water recharge for downstream rivers. Especially in arid regions where rainfall is scarce, meltwater greatly affects the year-round needs for downstream irrigation, domestic water, and industrial processing [3–6]. The Tianshan Mountains are the largest mountain range in Central Asia and are known as the "Water Tower of Central Asia". They are located in the hinterland of the Eurasian continent and have a wide distribution of snow and glaciers in the mountainous region. About 98% of the rivers originate in the mountainous region [7–9]. The snowmelt water accounts for 20–60% of the rivers and provides important river recharge for the arid regions of Central Asia. With global warming, the snow and glacier coverage in the northern hemisphere has begun

to gradually shrink, the snow period has been shortened, the snowmelt period has been early, and the frequency of spring floods has increased [10–12]. In recent years, the alpine snow area in the Tianshan region has decreased by 26.8–36.7% due to global warming [13], and the trend of shrinking is basically consistent with the worldwide snow accumulation change [14,15], but there are still differences in some partial mountainous areas. The change in snow cover leads to the change in hydrological characteristics of snowmelt runoff. Therefore, in order to utilize the snowmelt runoff resources more effectively and reduce the loss of life and property caused by floods, it is important to carry out the simulation study of snowmelt runoff in the Tianshan region.

In recent years, many scholars have carried out research on snowmelt runoff in the Tianshan Mountains. Shen, Y.-J. et al. [16] combined station data with APHRODITE data through the Mann–Kendall test and found that the snowmelt runoff in the Toxkon, Kumalik, Kaidu, and Huangshuigou basins, in the southern foothills of Tianshan, significantly increased in spring and winter. Deng, H. et al. [17] integrated literature, remote sensing data, and runoff data to analyze the glacial snow water resources in the middle Tianshan region. They have found that glacier thickness is decreasing and snow meltwater is increasing, which leads to the increase in river flow in this area, among which Aksu River has the most significant increase. Chen, H. et al. [18] used a large amount of hydrological, meteorological, and glacial data to quantify the difference in the contribution to a runoff between glacier and snowmelt for northern and central Tianshan. The results showed that the contribution of snow accumulation in northern Tianshan is 36%, which is 31% higher than the contribution of the glacier. With the rapid development of hydrological models [19], the numerical simulation of snowmelt runoff in mountainous areas has received great attention [20,21], and scholars in China have achieved valuable results in snowmelt runoff simulation experiments in the Tianshan region. Liu, Y. et al. [22] used an improved energy balance model, namely "UEBGrid", to simulate the proportion of rainfall, glacial meltwater, and snow meltwater, in the three typical spring flood formations in the Manas River on the northern slope of the Tianshan Mountains. The results showed that the floods are mainly caused by snow meltwater in low-altitude mountains, which accounts for 90% of the total runoff. Wang, X. et al. [23] also conducted runoff simulations in the Manas River Basin using an improved HBV model and concluded that from April to June, the runoff recharge in the Manas River is mainly snowmelt. Zhao, Q. et al. [24] found that, compared with the traditional VIC, the VIC coupled with the glacier ablation model has higher accuracy. In addition, affected by climate warming, the ablation rate of snow and ice in the Toxkan and Kunma Like basins, which are mainly recharged by snow and ice meltwater, increased, the ablation period was extended, and the snowmelt runoff in spring and autumn increased significantly. Ma, H. and Cheng, G. [25], earlier, applied the Snowmelt Runoff Model to mountainous areas in northwest China and performed a runoff simulation in the Gongnaisi River Basin of western Tianshan Mountains as the study area. The results confirmed the applicability of the model in the watershed, while the occurrence of snowmelt runoff would be advanced to early spring under the assumed 4 °C temperature increase scenario.

The sparse distribution of ground stations and the scarcity of data information are major challenges for the simulation of snowmelt runoff in mountainous areas [26]. With the development of remote sensing technology, remote sensing data with high spatial and temporal resolution can make up for the lack of ground data [27]. Among the hydrological models, the Snowmelt Runoff Model (SRM) has been widely used to simulate snowmelt runoff in mountainous areas. The SRM is a conceptual hydrological model, based on the degree-day factor [28,29]. It is also one of the typical hydrological models that use remotely sensed data as the main input variable [30]. Therefore, it is more suitable for simulating snowmelt runoff in mountainous areas, where information is scarce.

MOD10A2 is an 8-day snow cover composite product that is a composite of 2 to 8 daily snow cover products from the Moderate Resolution Imaging Spectroradiometer [31]. MOD10A2 is the most commonly used remote sensing data for snow cover area (SCA) and has been used in many mountain snowmelt runoff simulation studies [32,33]. Some scholars

also used the remote sensing data of Landsat 7 and MOD10A2 to retrieve snow areas to drive SRM and achieved good results [34]. In the last five years (2017–2021), research on SRM had improved the availability and simulation accuracy of the SRM in areas where data is scarce. On the other hand, it highlighted the ability of SRM to predict snow and ice meltwater runoff under climate change. Javeria Saleem et al. [35] performed SRM in the Hunza watershed and the results showed that regional warming affects local hydrological characteristics. Caitriona Steele et al. [36] compared the accuracy of the application of snow cover product from MODIS and Landsat TM in the SRM, and the two products were in high agreement in terms of snow cover area, but Landsat TM snow cover product MODSCAG was more suitable for the SRM when the study watershed area was less than 4000 km$^2$. Huma Hayat et al. [37] used the Digital Elevation Model (DEM) of Japanese Advanced Land Observation Satellite (ALOS), with 12.5 m spatial resolution, to analyze the accuracy of the runoff simulation by basin and zones. The results showed that the accuracy of the basin was higher in the same study region. Muhammad Adnan et al. [38] predicted changes in temperature and rainfall in the Gilgit River Basin using the regional climate model PRECIS, and the runoff would increase by 35–40% under the scenario of a 3 °C increase in the average temperature of the year. Meanwhile, the expansion of snow and glacier area due to greater precipitation would also increase the runoff volume. Muhammad Azmat et al. [30] carried out future runoff simulation prediction using SRM, HEC-HMS, and the increase in the runoff for the former scenario was less than that of the latter under higher temperature and greater precipitation. All these findings will contribute to the research on the hydrology of the cryosphere under climate change.

In 2020, the National Cryosphere Desert Data Center in China released the first version of the "China MODIS daily cloud-free 500 m snow area product dataset" (MODIS_CGF_SCE V01) [39], which has a higher temporal resolution than MOD10A2 and better meets the needs of snow variables in SRM. However, there are very few applications that use this dataset in SRM, and little is known about whether the dataset is more suitable to obtain better results in SRM. Therefore, in this paper, we used the MODIS_CGF_SCE as the data source to calculate the snow area and carry out the snowmelt runoff simulation experiments in the Hutubi River Basin, in the middle part of the northern slope of the Tianshan Mountains. The results of the simulation were taken to evaluate the applicability of the SRM in the Hutubi River Basin.

## 2. Materials and Methods

### 2.1. Study Area

The Hutubi River Basin is located in the middle of the northern slope of the Tianshan Mountains in China (86°05′–87°08′ N, 43°07′–45°20′ E) and is about 258 km long from north to south and 40 km wide from east to west, with a total area of 10,255 km$^2$. The topography of the basin is higher in the south and lower in the north, with 35% mountains, 40% plain, and 25% desert distributed from south to north. The geological characteristics of the plains are characterized by the transformation from pebbles and gravels to gravelly coarse sand, medium-coarse sand, and fine sand. The irrigated agricultural area is distributed in the plain area with an area of 867 km$^2$ [40]. The Hutubi River Basin has a continental climate, with hot summer and cold winter. The area is abundantly illuminated with a multi-year average temperature of 6.7 °C, average precipitation of 171.2 mm, and the annual runoff is about $4.7 \times 10^8$ m$^3$. The basin is rich in snow and ice resources, with an annual snowmelt volume of about $0.524 \times 10^8$ m$^3$ [41]. In the spring, the runoff recharge is mainly snow meltwater, and more than 50% of water resources in the irrigation area come from snow meltwater [42]. The study area of this paper is the catchment area above the Shimen hydrological station of Hutubi River, with an area of 1840 km$^2$, a mean elevation of 2984 m, and a river longitudinal drop ratio of 23.13%. This area is the main area of runoff formation, the river is 88 km long, the total surface flow accounts for 93.6% of the whole river, and the interannual variation is small [43]. Thus, this area is unique and representative in the simulation of snowmelt runoff in the Tianshan region. The study area is shown in Figure 1.

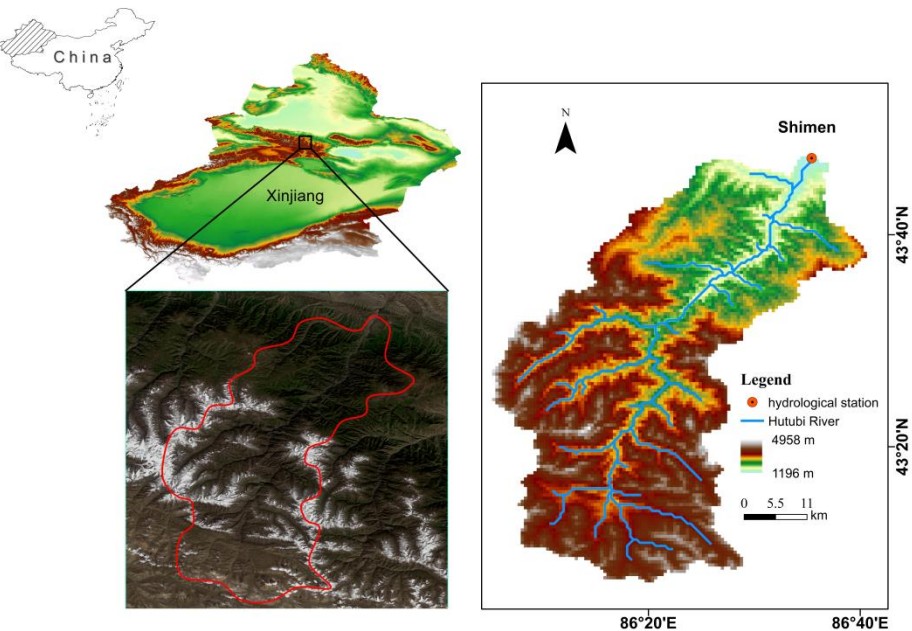

**Figure 1.** Schematic diagram of the study area.

*2.2. Materials*

The data used in this paper include remotely sensed data, meteorological data, and hydrological data. The remotely sensed data are Digital Elevation Model (DEM) data and snow product data. The DEM data, which is the basis of terrain analysis in the study area, was derived from ASTER GDEM V2 (https://www.gscloud.cn, accessed on 1 January 2020). The DEM data was developed and improved with the participation of NASA and had a spatial resolution of 30 m. The snow data is "MODIS Daily Cloud-free Snow Cover 500 m Dataset over China" (MODIS_CGF_SCE) (http://www.crensed.ac.cn/portal/, accessed on 1 January 2020), which was calculated based on MODIS day-by-day surface reflectivity products MOD09GA and MYD09GA [44] with a spatial resolution of 500 m and a daily temporal resolution. The data cover all regions of China. The meteorological data and hydrological data were obtained from the Hutubi River Shimen hydrological station, including daily precipitation, average daily temperature, and average daily flow. The time range of the data is 2003–2009.

*2.3. SRM and Parameter Determination*

2.3.1. Model Structure

The SRM is a conceptual hydrological model based on the degree-day factor developed by J. Martinec [28]. The initial experiments were carried out in a small watershed of 43.3 km$^2$ in Europe. With the development of remote sensing technology, the acquisition of snow areas by satellite has made the application of SRM more extensive, and the maximum watershed area has now reached $9.17 \times 10^5$ km$^2$ with a maximum elevation of 8840 m [45]. The model converts the daily snowmelt and rainfall in different elevation range into the average daily flow, which can be calculated by Equation (1):

$$Q_{n+1} = [C_S \cdot a \cdot (T_n + \triangle T) \cdot S_n + C_r \cdot P_n] \cdot \frac{A \cdot 10,000}{86,400} \cdot (1 - k_{n+1}) + Q_n \cdot k_{n+1} \quad (1)$$

where Q represents the average daily flow (m$^3 \cdot$s$^{-1}$), $C_S$ is the snowmelt runoff coefficient, $C_r$ is the rainfall-runoff coefficient, a represents the degree-day factor (cm$\cdot°$C$^{-1}\cdot$d$^{-1}$), T represents the number of degree-day factors ($°$C$\cdot$d), $\triangle$T represents the modified value of the degree-day factor ($°$C$\cdot$d), S is the ratio of the snow cover area to the total area, P is the daily precipitation (cm), A is the watershed area (km$^2$), 10,000/86,400 converts units to m$^3 \cdot$s$^{-1}$, k represents the runoff recession coefficient, and n is the sequence of days during

the discharge computation period. In addition, based on climate trends, the SRM can predict forward runoff based on the rainfall, temperature, and snow characteristics of the predicted scenarios. The structure of the SRM is illustrated in Figure 2.

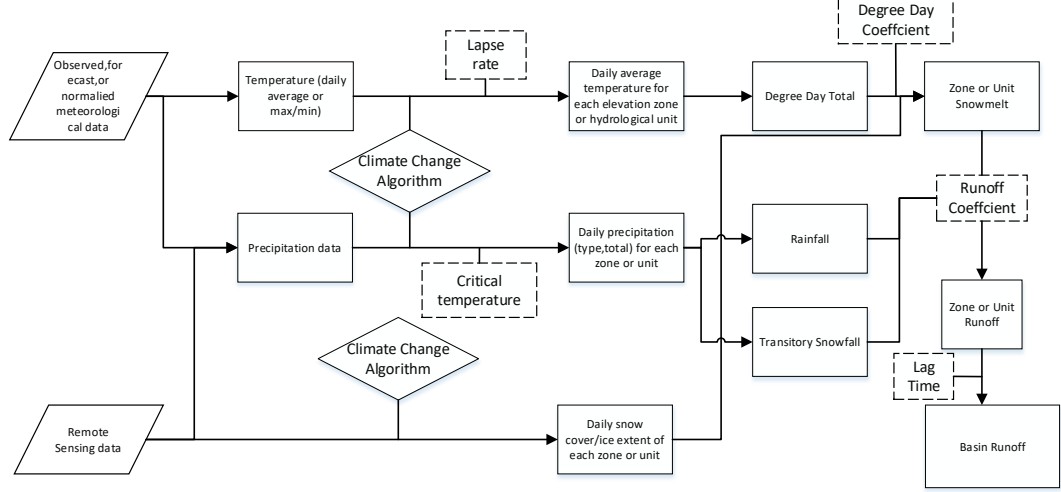

**Figure 2.** Working principle and structure of SRM.

The simulation accuracy of SRM is generally evaluated by two coefficients, i.e., the coefficient of determination ($R^2$) and the volume difference ($D_V$), which can be calculated by the following equations:

$$R^2 = 1 - \sum_{i=1}^{n} (Q_i - Q_i')^2 / \sum_{i=1}^{n} (Q_i - \overline{Q})^2 \tag{2}$$

$$D_V = 100 \cdot (V_R - V_R') / V_R \tag{3}$$

where $Q_i$ and $Q_i'$ are the measured and simulated average daily flows, respectively; $\overline{Q}$ is the simulated period average flows, n is the simulated time series, and $V_R$ and $V_R'$ are the total measured and simulated runoff ($m^3$), respectively. $R^2$ is in the range of 0–1. The closer the value of $R^2$ to 1, the higher the simulation accuracy; the closer the absolute value of $D_V$ to 0, the higher the simulation accuracy. In addition, the simulation accuracy was also evaluated by Pearson correlation coefficient r.

### 2.3.2. Elevation Zone

Elevation zoning is the basic treatment before the model runs, and the study watershed is divided into several sub-regions according to the elevation intervals. In this paper, the study area was divided into 7 elevation zones with a 500-m elevation interval, and the area of each elevation zone was calculated separately. The elevation at one-half of the area is determined, which is referred to as the mean elevation. The mean elevation is an important parameter of the model input variables. The distribution of elevation zones in the study area is shown in Figure 3, and Table 1 shows the information of each elevation zone.

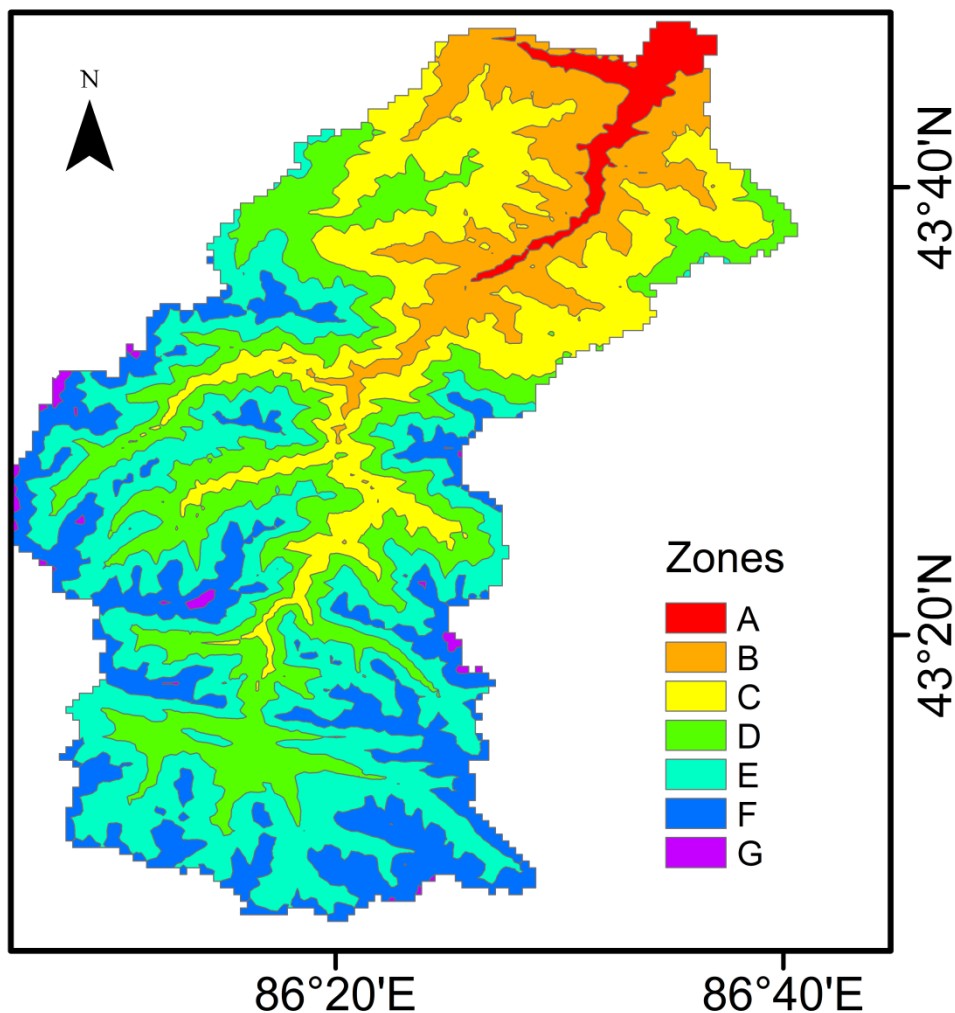

**Figure 3.** Schematic diagram of elevation zoning in the study area.

**Table 1.** A–G elevation zones information sheet.

| Zone | Elevation Range/m | Average Elevation/m | Area/km$^2$ |
|------|-------------------|---------------------|-------------|
| A | 1196–1696 | 1565 | 57 |
| B | 1696–2196 | 1864 | 223 |
| C | 2196–2696 | 2450 | 390.25 |
| D | 2696–3196 | 2960 | 439.75 |
| E | 3196–3696 | 3542 | 583 |
| F | 3696–4196 | 3850 | 324.25 |
| G | 4196–4696 | 4292 | 12.5 |

### 2.3.3. Model Variables

Daily snow cover area (SCA), average daily temperature, and daily precipitation are the three input variables of SRM. SCA is the ratio of snow area to the total area in an elevation zone. The snow area was obtained from the MODIS_CGF_SCE and divided into six types of features. As shown in Table 2, each image value represents a land type.

**Table 2.** MODIS_CGF_SCE pixel digital number and representative land type.

| Pixel Digital Number | Land Type |
|:---:|:---:|
| 0 | Land |
| 1 | snow cover |
| 2 | Snow water equivalent interpolation of snow |
| 3 | Inland water or ocean |
| 4 | Glacier |
| 255 | Filling value |

Since there is no experiment on the application of MODIS_CGF_SCE in the SRM model, this simulation extracted the pixel digital number with the values of 1, 2, and 4 as the snow-covered surface types and combined them with the elevation zone to determine the snow coverage rate. The temperature and precipitation data were obtained from the Shimen hydrological station. The temperature and precipitation at the mean elevation were used as input to the SRM model. Thus, when the station elevation disagreed with the mean elevation of the elevation zone, the temperature and precipitation at the mean elevation were determined by the derivation of station values. The temperature is generally calculated according to the formula for the direct temperature drop rate.

$$T = \overline{T} + \triangle T \tag{4}$$

$$\triangle T = \gamma(h_{st} - \overline{h})/100 \tag{5}$$

where T is the average daily temperature at the mean elevation (°C·d), $\overline{T}$ is the average daily temperature at the station, $\triangle T$ is the temperature correction value, $\gamma$ is the direct temperature drop rate, which is taken as 0.3 °C/100 m, 0.67 °C/100 m, and 0.26 °C/100 m, respectively, depending on the season [46]. $h_{st}$ is the elevation where the station is located, and $\overline{h}$ is the mean elevation of the elevation zone. The derivation of precipitation is similar to that of temperature. In general, precipitation in mountainous areas increases with elevation. In this paper, we refer to the results of Ji, X. and Chen, Y. [47], and adopt different precipitation change rates in three elevation ranges (<2500 m; 2500–3700 m; >3700 m).

2.3.4. Model Parameters

The main parameters of the model include degree-day factor a (cm·°C$^{-1}$·d$^{-1}$), runoff coefficient C, critical temperature $T_{CRIT}$ (°C), runoff recession coefficient k, lag time L (h), and temperature lapse rate $\gamma$. The degree-day factor indicates the depth of snowmelt per 1 °C increase in temperature and is often expressed by the empirical formula between snow density and water density.

$$a = 1.1\rho_s/\rho_w \tag{6}$$

where $\rho_S$ refers to the snow density, and $\rho_W$ is the water density. The runoff coefficient includes snowmelt runoff coefficient ($C_S$) and rainfall-runoff coefficient ($C_r$), which represent the contribution of snowmelt and rainfall to runoff, respectively. The critical temperature $T_{CRIT}$ is used to determine the precipitation pattern. When the average daily temperature is higher than the critical temperature, the precipitation is considered rainfall and is controlled by the rainfall-runoff coefficient, which is directly applied to the runoff simulation. On the contrary, when the average daily temperature is lower than the critical temperature, the precipitation is considered snowfall and is stored on the surface, which is calculated as snow meltwater when the temperature rises. Usually, the critical temperature value is larger at the beginning of snowmelt and smaller at the end. Runoff recession factor k represents the natural decrease in the percentage of runoff volume in the absence of rainfall or snowmelt to recharge runoff. The value of k is defined as the ratio of two adjacent days of runoff data, as shown in the following equation:

$$k_n = Q_{n+1}/Q_n \tag{7}$$

where $Q_n$ and $Q_{n+1}$ are the average daily flow of the current day and the next day in the unit of $m^3 \cdot s^{-1}$, respectively. Since the runoff rate changes from time to time, the values of k can be further expressed as

$$k_{n+1} = x \cdot Q_n^{-y} \tag{8}$$

where x and y are constants, whose values can be obtained by logarithmical solving:

$$\log k_1 = \log x - y \log Q_1 \tag{9}$$

$$\log k_2 = \log x - y \log Q_2 \tag{10}$$

Then, the values of x, y, and k under the time series can be determined. L refers to the delay time for the recharge source to reach the hydrographic cross-section. Equations (1)–(10) were derived from the research of J. Martinec et al. [45]. The study area of this paper is the Hutubi River Basin where no scholars have worked on SRM research, and there is a lack of snowpack and meteorological data. Therefore, in this paper, we referred to the results of the research on SRM models in Xinjiang to determine the initial parameter range and then conducted simulation experiments with 3 years of data through the empirical regression analysis, with the final parameter scheme determined when the results reached the optimum [48]. The parameters $\gamma$ and k have been proposed as calibration methods in the previous section, and the a, C, $T_{CRIT}$, and L are the main calibrated parameters in this paper. Statistically, the range of values of a, C, $T_{CRIT}$, and L [25,49–52] for research on SRM in Xinjiang mountainous region are shown in Table 3.

**Table 3.** Initial parameter range.

| Parameter | Value Range |
| --- | --- |
| a | 0.15–0.36 |
| $C_S$ | 0.2–0.73 |
| $C_r$ | 0.05–0.9 |
| $T_{CRIT}$ | 0–3 |
| L | 3–18 |

Table 3 was applied as the initial parameter substituted to the database of 2003–2005 for runoff simulation, and the parameters of SRM were calibrated according to the method of Martince et al. [44]. Martince et al. suggested that the degree-day factor was a value that varied with the snowmelt season and was non-constant, with illumination and wind speed affecting the value. It should be noted that when glaciers are involved in the study area, a is usually greater than 0.6 $cm \cdot {}^{\circ}C^{-1} \cdot d^{-1}$, which decreases in the advanced period of snowmelt, under the influence of snowfall. The C is larger in the early stage of snowmelt and decreases in the later stage. It is due to the insignificant loss of recharge of runoff from snowmelt and precipitation in the early stage. As the snowmelt process continues, vegetation and soil are exposed on the ground surface, which retains part of the surface runoff, thus causing the C to be decreased. During simulation months of May–October, the trend of temperature change during this period increases and then decreases, and the rainfall mainly occurs in June–August, so the C is higher in June–August and lowers in May, September, and October. For abnormal weather such as heavy rainfall, the temperature suddenly declines and then rises steeply, and the daily runoff coefficients need to be specifically adjusted. With heavy rainfall, the $C_r$ would need to be adjusted downward suitably; with a steep rise in temperature, the $C_s$ need to be increased. This situation was observed from May to October in 2003–2005. The a of this calibration result was different compared with Table 3. Considering the presence of glaciers in the Hutubi River Basin, the runoff simulation results with a maximum a value of 0.36 $cm \cdot {}^{\circ}C^{-1} \cdot d^{-1}$ did not achieve the best accuracy, but when the value was increased to 0.5 $cm \cdot {}^{\circ}C^{-1} \cdot d^{-1}$, the simulation results were closer to the measured data. The critical temperature is basically consistent with Table 3. The precipitation pattern at high temperatures is mainly rainfall, so the $T_{crit}$ is

lower in the high-temperature season [52]. L has a correlation with the size of the watershed area, referring to Martinec's research, the 2170 km$^2$ runoff lag time is 12.5 h, our study area is 2029.5 km$^2$, so the value is taken as 12 h in this paper. Parameter calibration is a cyclical process, and the results of this simulation were finally determined after several adjustments as shown in Table 4.

**Table 4.** Variation range/value of each parameter of SRM.

| Month | a | $C_S$ | $C_r$ | $T_{CRIT}$ | x | y | $\gamma$ |
|---|---|---|---|---|---|---|---|
| May | 0.08–0.2 | 0.1–0.5 | 0.1–0.15 | 2 | | | 0.3 |
| June | 0.2–0.35 | 0.5–0.8 | 0.15–0.7 | 1.5 | | | |
| July | 0.2–0.35 | 0.8–1 | 0.2–0.6 | 0.5 | 1.02 | 0.1 | 0.67 |
| August | 0.35–0.5 | 0.58–1 | 0.2–0.5 | 0.5 | | | |
| September | 0.28–0.5 | 0.3–0.5 | 0.1–0.5 | 1.5 | | | 0.23 |
| October | 0.35 | 0.3 | 0.1 | 1.5 | | | |

## 3. Results

### 3.1. Snow Cover Characteristics

By analyzing the annual trend of SCA in each elevation zone, it can be observed that the zero-snow cover occurred continuously in the A–D elevation zones (1196–3196 m), between July and August, so the area below 3200 m elevation was classified as seasonal snow area (1110 km$^2$), and the area above 3200 m as permanent snow area (919.75 km$^2$) in this paper. The seasonal snow area accounts for more than half of the total area of the study area. According to the change curves of SCA, between the seasonal snow area and permanent snow area in the Hutubi watershed from 2003 to 2009 (Figure 4), the following observations can be obtained: (1) The trends of SCA of the seasonal snow area (black solid line) and permanent snow area (red solid line) are generally the same, and the SCA of the permanent snow area is usually larger than that of the seasonal snow area. (2) The seasonal snow starts to accumulate gradually in October, and the SCA is less than 0.1 from October to early November. The SCA increases significantly from mid-November, and the main period of seasonal snow distribution is from mid-November to February, with a maximum annual mean SCA of 0.3. The seasonal snow gradually melts in March, and the melting process ends at the end of May. (3) In summer (July–August), the snow cover in the study area is minimal, and the minimum annual mean SCA in the permanent snow area is 0.007. Compared with the seasonal snow area, the SCA in the permanent snow area increases significantly in early October, with an SCA of 0.34. The SCA in the permanent snow area increases again in early November, and the main period of permanent snow area distribution is from mid-November to mid-May, with high snow cover lasting for 6 months and a maximum annual mean SCA of 0.46. The maximum average annual SCA is 0.46. The ablation process in the permanent snow area mainly occurs from late May to early July.

### 3.2. Runoff Characteristics

The measured average daily flow (black solid line) and the simulated average daily flow (red solid line) of the Hutubi River from 2003 to 2009 are shown in Figure 5. From the figure, 2003–2005 is the rate period and 2006–2009 is the validation period. From May to October, the flow of the Hutubi River first increases and then decreases. The flood peak is concentrated from June to August, which is the high-water period in the Hutubi River. The maximum average daily flow from 2003 to 2009 was 171 m$^3$·s$^{-1}$, which occurred on 10 July 2007. May, September, and October are normal water periods, with relatively stable variations in the average daily flow.

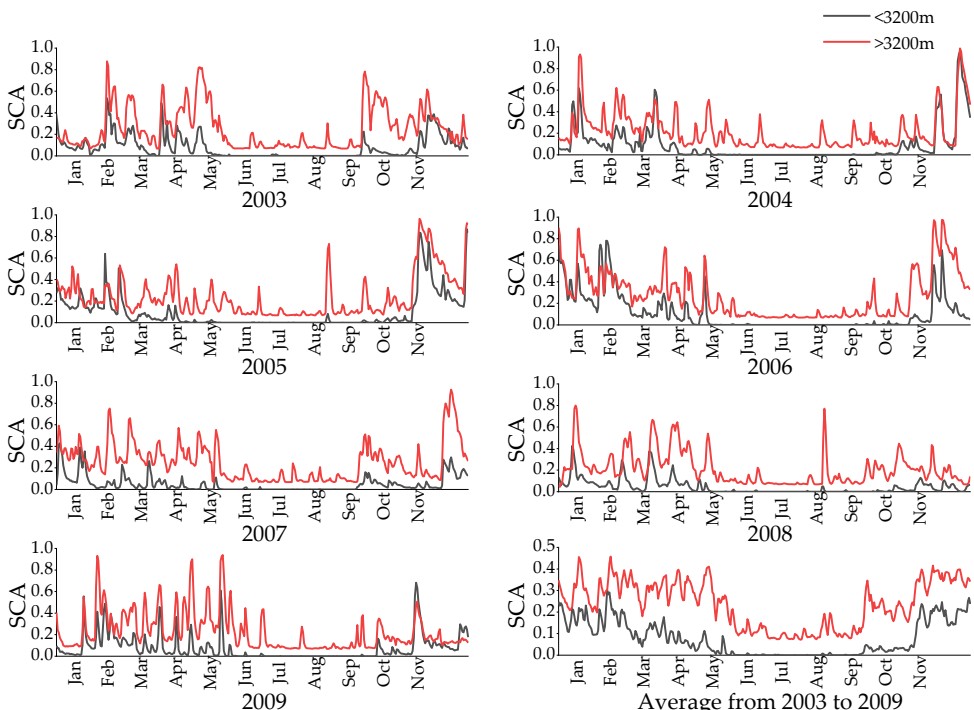

**Figure 4.** Characteristics of SCA in the Hutubi River Basin.

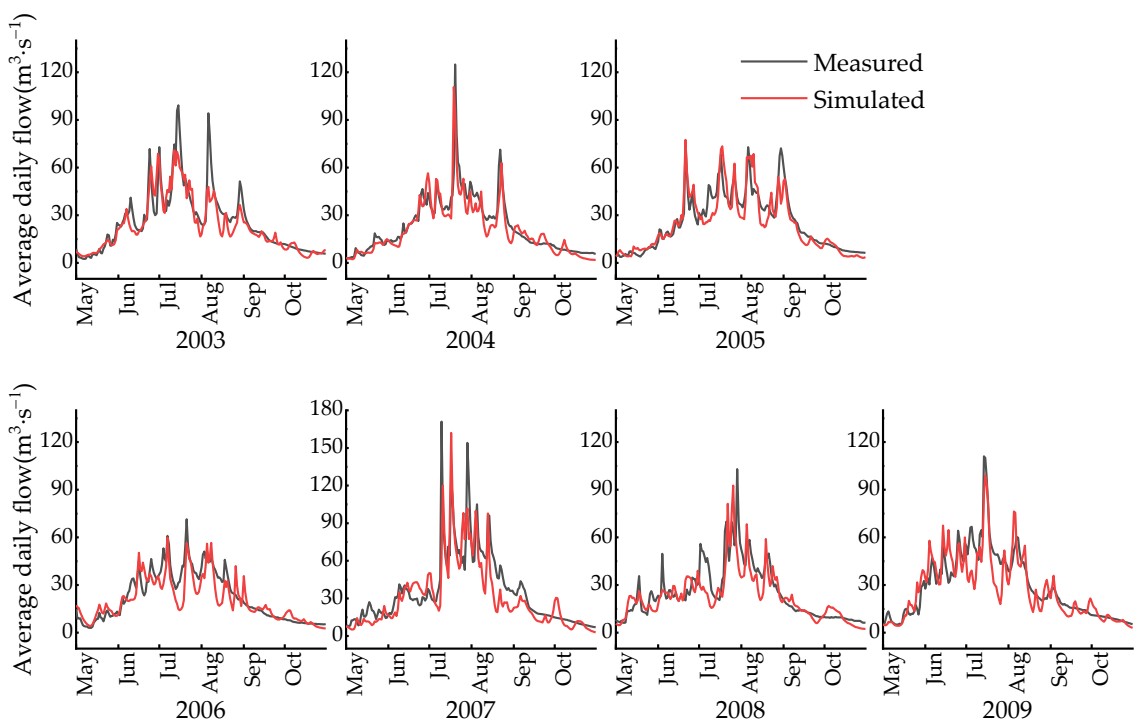

**Figure 5.** Average daily flow change curve in the Hutubi River from 2003 to 2009.

Figure 6 shows the characteristics of the statistical distribution of the average daily flow by month from May to October in 2003–2009. The average monthly flow rate in July is the largest, and the measured and simulated values are 50.9 $m^3 \cdot s^{-1}$ and 44.4 $m^3 \cdot s^{-1}$, respectively. The measured values range from 30.5 $m^3 \cdot s^{-1}$ to 74.7 $m^3 \cdot s^{-1}$ and the simulated values range from 14.2 $m^3 \cdot s^{-1}$ to 92.7 $m^3 \cdot s^{-1}$; thus, the simulated values fluctuate more than the measured values in the distribution range of 10–90%. The average monthly flow is the smallest in October, and the measured and simulated values are 8.4 $m^3 \cdot s^{-1}$

and 7.9 $m^3 \cdot s^{-1}$, respectively. The measured flow varies from 6.0 $m^3 \cdot s^{-1}$ to 11.2 $m^3 \cdot s^{-1}$ in the distribution range of 10–90%, while the simulated values vary from 1.9 $m^3 \cdot s^{-1}$ to 19.5 $m^3 \cdot s^{-1}$, with a larger fluctuation range than the measured values.

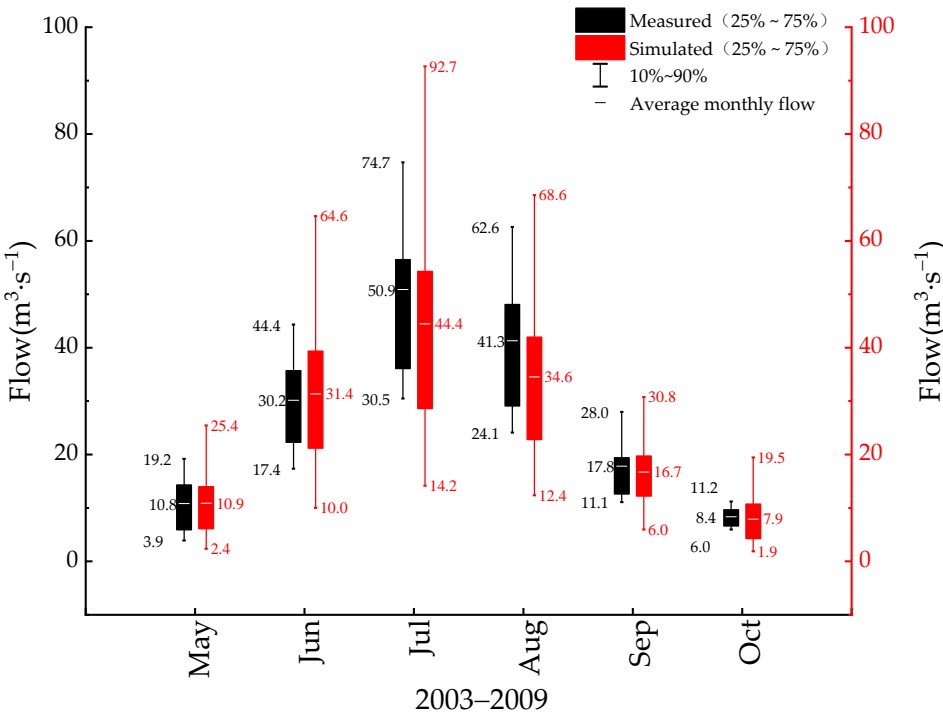

**Figure 6.** Distribution characteristics of average daily flow in 2003–2009.

Figure 7 shows the characteristics of monthly runoff and total flow from May to October in 2003–2009. The maximum values of the measured monthly runoff in May, July, August, September, and October were observed in 2007, which were $4.21 \times 10^7$ $m^3$, $1.91 \times 10^8$ $m^3$, $1.62 \times 10^8$ $m^3$, $6.39 \times 10^7$ $m^3$, and $2.92 \times 10^7$ $m^3$, respectively. The total measured flow from May to October in 2003–2009 was $4.10 \times 10^8$ $m^3$, $3.66 \times 10^8$ $m^3$, $4.05 \times 10^8$ $m^3$, $3.57 \times 10^8$ $m^3$, $5.66 \times 10^8$ $m^3$, $3.99 \times 10^8$ $m^3$, and $4.58 \times 10^8$ $m^3$, respectively, with a generally increasing trend. The total flow from May to October in 2007 was the largest.

The simulated monthly flow results from 2003 to 2009 are consistent with the measured values. The simulated values are smaller than the measured values in May of 2004, 2007 and 2008, June of 2003, 2004 and 2006, and September of 2003, 2005, 2007 and 2009. The difference between the simulated and measured values is most prominent in July and August of 2003 to 2009, and the simulated values are smaller than the measured values in July and August of 2003 to 2009. The simulated values in October of 2003, 2004, and 2005 are smaller than the measured values, and the differences between the simulated and measured values in October of 2007, 2008, and 2009 are small, which are 2.2%, 2.0%, and 0.5% of the measured values, respectively. The total flow variation from May to October in 2003–2009 shows that the simulated value of each year is smaller than the measured value. In summary, the simulated value is generally smaller than the measured value.

### 3.3. Simulation Accuracy Evaluation

The model coefficient of determination ($R^2$) and the volume difference ($D_V$) from 2003 to 2009 are shown in Table 5. According to the results of the snowmelt runoff simulation accuracy comparison, conducted by the World Meteorological Organization (WMO) in 1986, the mean volume difference ($\overline{D_V}$) of the SRM model was 5.97% and the mean coefficient of determination ($\overline{R^2}$) was 0.81 in the test basins. The $R^2$ of the calibration period outperforms the WMO statistics. The $D_V$ values of 2003 and 2004 are 9.81% and 8.55%, respectively,

which are both greater than 5.97%, and the $D_V$ value of 2005 is 4.48%, which is less than 5.97%. The overall simulation accuracy of 2005 is the best of the calibration period. Overall, the simulation accuracy of the calibration period is better. The simulation accuracy of the validation period is lower than that of the calibration period, and the best simulation accuracy is in 2009, with $R^2$ and $D_V$ of 5.07% and 0.77, respectively. The mean value of $R^2$ in the validation period is 0.73, and the mean value of $D_V$ is 8.85. Statistically, the mean accuracy coefficients of $R^2$ and $D_V$ for the validation period of SRM in other watersheds of the Tianshan region in China (Yarkant River, Kashi River, Urumqi River, Manas River, Kuitun River, and Tashikuergan River [49,53–57]) in the last decade are 0.79 and 6.24%, respectively. This shows that the accuracy of this SRM simulation in the Hutubi River Basin is at the same level as the overall accuracy of rivers in the Tianshan region, which confirms the adaptability of SRM for the snowmelt runoff simulation in the Hutubi River Basin.

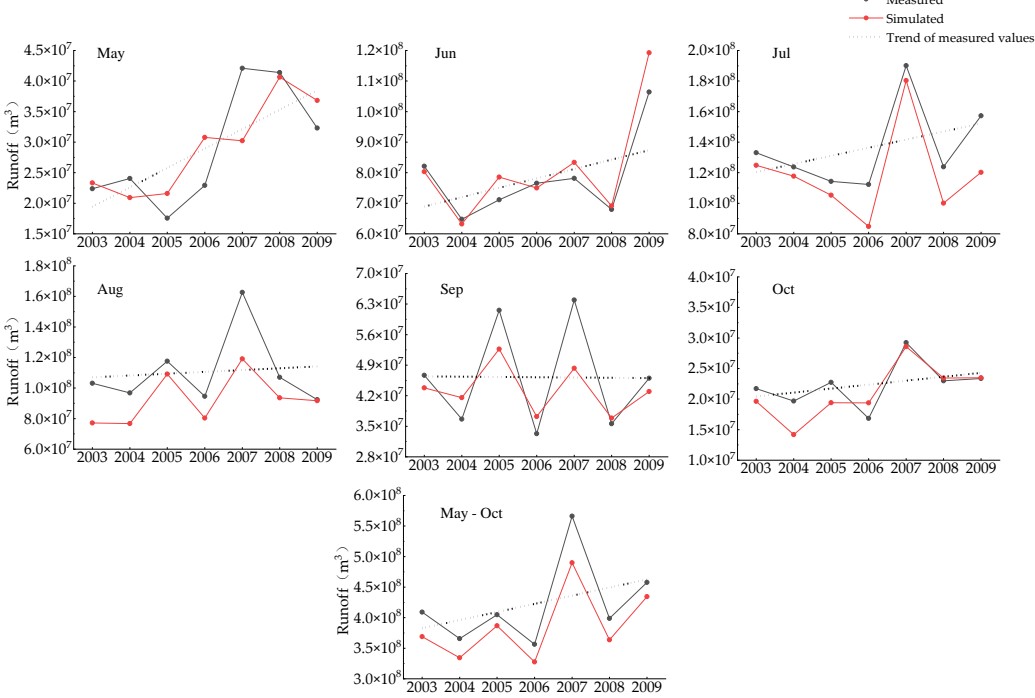

**Figure 7.** Variation curve of monthly runoff volume in 2003–2009.

**Table 5.** Model accuracy coefficients in 2003–2009.

|  | Time | $D_V$ | $R^2$ |
|---|---|---|---|
| Calibration period | 2003 | 9.81% | 0.81 |
|  | 2004 | 8.55% | 0.82 |
|  | 2005 | 4.48% | 0.83 |
| Validation period | 2006 | 8.13% | 0.76 |
|  | 2007 | 13.46% | 0.71 |
|  | 2008 | 8.75% | 0.66 |
|  | 2009 | 5.07% | 0.77 |

The Pearson correlation coefficients r between the measured and simulated average daily flows during the normal water period (May, September, and October) and the high-water period (June, July, and August) were compared. The results showed that the correlation between the measured and simulated values was higher and the runoff simulation was better during the normal water period. The correlations between the normal water period and the high-water period, from 2003 to 2009, are shown in Figure 8. From

the figure, it can be seen that the value of r in the normal water period is 0.83, and the value of r in the high-water period is 0.77.

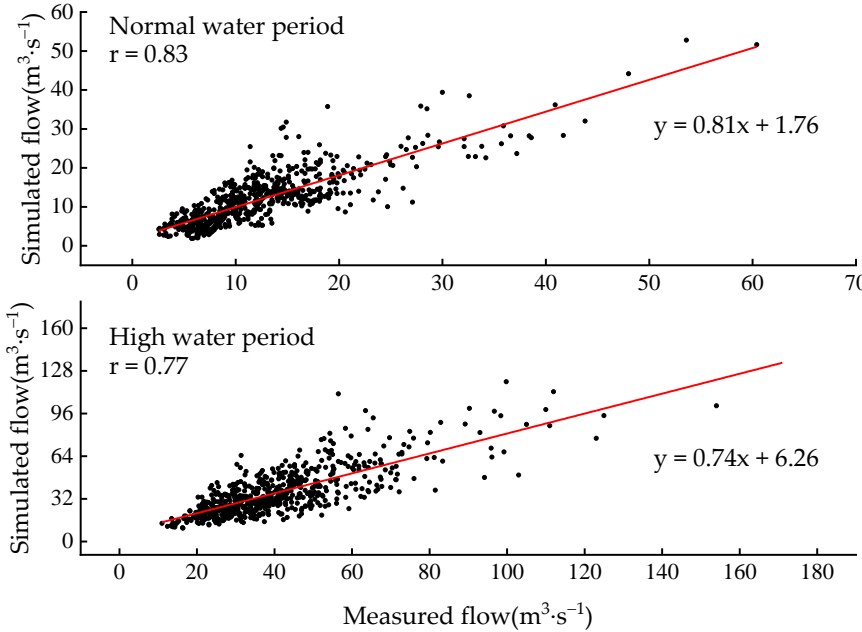

**Figure 8.** Correlation between the measured and simulated values in the Hutubi River Basin in 2003–2009.

## 4. Discussion

### 4.1. Impact of Extreme Weather Events on Runoff

With global climate change, the general trend of annual precipitation and temperature in the Xinjiang region is increasing [58], and the frequency of extreme precipitation and extreme temperature (high and low temperature) events is increasing, affecting the characteristics of river flow.

Using percentiles [59], extreme precipitation events from May to October (2003–2009) were classified with the 95% quantile as the threshold, and daily precipitation events greater than 17.2 mm were judged as extreme precipitation events. Extremely high temperature and extremely low temperature were classified using the 95% and 5% quantile as the threshold. The events higher than 24.1 °C were judged as extremely high temperature and those lower than 4.2 °C were judged as extremely low temperature. From May to October, there were 21 extreme precipitation events, among which 7 extreme precipitation events occurred in May, with a probability of 33%, 5 extreme precipitation events occurred in June, and 6 extreme precipitation events occurred in July. The probability of extreme precipitation events occurring in August and September was low, and there was only one extreme precipitation event in September, and no extreme precipitation event in October. May had the highest frequency of extreme precipitation events, but its direct impact on runoff was relatively minor and there was no significant growth of flow, which may be explained by the fact that May was in a period of fluctuating temperature rise, with lower average temperatures and even extreme cold weather. When the temperature is low, precipitation is stored on the surface, in the form of snow, and continues to recharge the runoff as the temperature rises. For example, the maximum extreme precipitation of the simulated period from 2003 to 2009 occurred on 25 May 2009, with 42.5 mm of precipitation and an average daily flow of 17.4 $m^3 \cdot s^{-1}$. On the next day, the extremely low temperature was 4.1 °C and the average daily flow was 14.9 $m^3 \cdot s^{-1}$, which was less than the flow on May 25. Although the flow on May 25 and 26 did not increase significantly, the snow cover in the study area increased from 20% to 73%. The difference between the snow area in the study area on 25 and 26 May 2009 is shown in Figure 9.

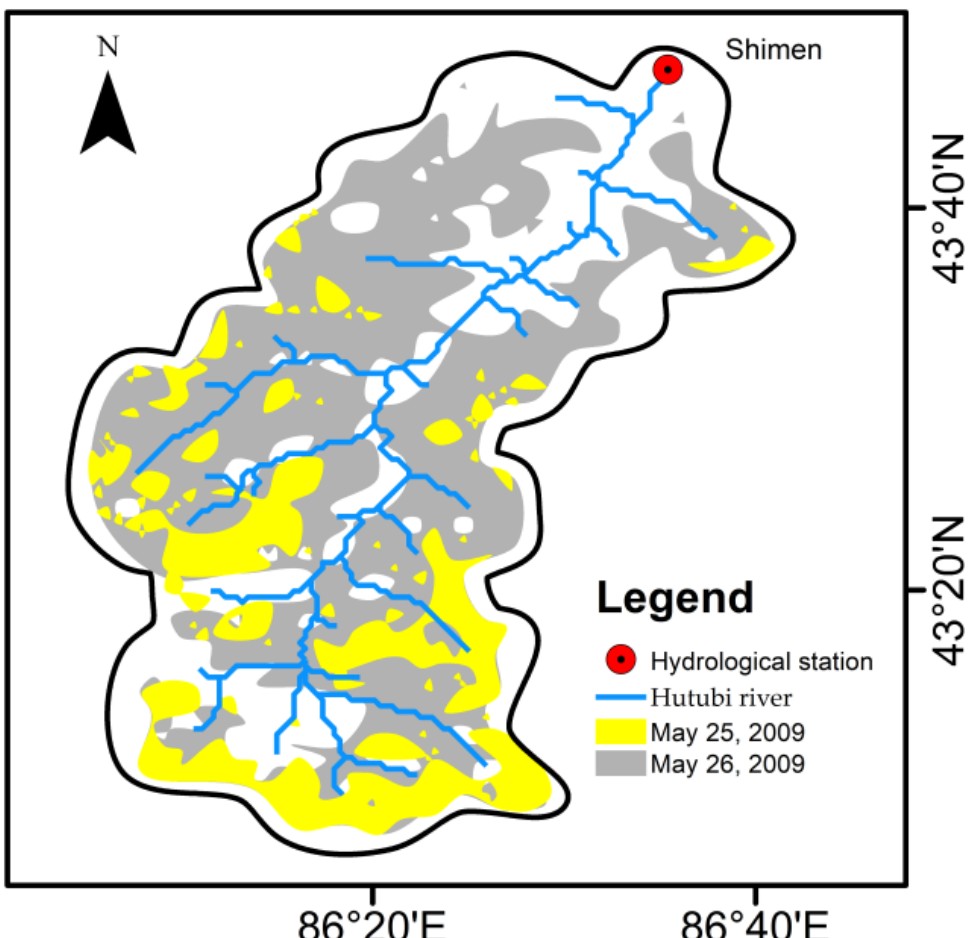

**Figure 9.** Change in snow area under maximum extreme events in May 2009 (accessed on 10 January 2020).

Extreme precipitation events from June to July are one of the most important factors influencing flood peak formation, which may form on the same day or the day after extreme precipitation occurs during that period. The most prominent flow variation was observed on 10 July 2007, when the extreme precipitation was 20 mm and the flow was 171 $m^3 \cdot s^{-1}$. Compared to the previous day's flow of 45.4 $m^3 \cdot s^{-1}$, the difference was 125.6 $m^3 \cdot s^{-1}$. In addition to extreme precipitation events, the occurrence of continuous non-extreme precipitation may also lead to the formation of flood peaks. For example, in 2007, precipitation occurred from 22 to 28 July, and the flow on 29 July was 154 $m^3 \cdot s^{-1}$, which was increased by 71.1 $m^3 \cdot s^{-1}$ from 82.9 $m^3 \cdot s^{-1}$ on 28 July. Figure 10 shows the average daily flow and precipitation variation in July 2007. The reason for the smaller effect of extremely high temperature on flow variation from June to July, compared to extreme precipitation, is that June and July are the periods of minimal snow cover and minimal snowmelt recharge. The probability of extreme precipitation events from August to September is small. Thus, the flood formation in August is related to continuous precipitation, and the flow gradually decreases, starting in September. October is the main period when extremely low temperature events occur, and precipitation starts to change from rainfall to snowfall, which, in turn, is stored as snow and rarely recharges to runoff.

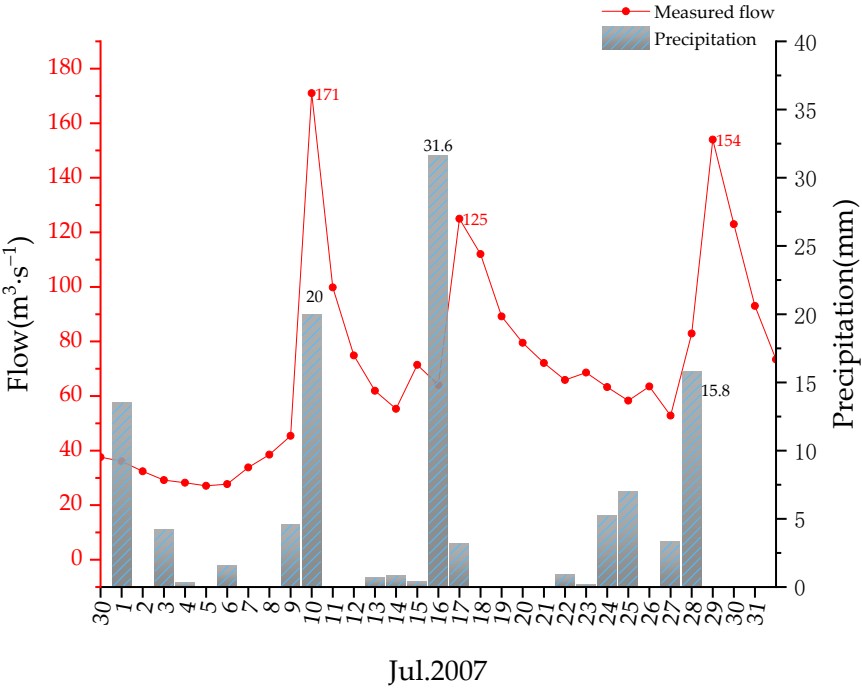

**Figure 10.** The average daily flow and precipitation variation in July 2007.

*4.2. Uncertainty Analysis and Optimization*

In this study, the simulation results are generally smaller than the measured values. The uncertainties in the hydrological model simulations arise from the model structure, input variables, and parameters. The SRM model establishes the relationship between snow, rainfall, and runoff, and achieves better simulation results in different watersheds. Although snow and rainfall are the main recharge for snowmelt runoff in mountainous areas, groundwater is also one of the sources of water recharge for rivers in the Tianshan region [60]. Therefore, the lack of groundwater recharge may cause the simulated values to be lower than the measured values. In contrast to the uncertainties arising from the model structure, researchers have focused on the uncertainties arising from the variables and parameters and studied their optimization methods.

Insufficient precipitation data is one of the factors contributing to the high uncertainty of SRM input variables. Therefore, when the actual precipitation at a high elevation is greater than that at a low elevation, using low elevation precipitation data as the overall input will underestimate the precipitation and lead to smaller simulation results than the measured values. To optimize this uncertainty, attempts have been made to improve the accuracy of precipitation variables by combining remotely sensed precipitation data to compensate for the lack of site data. For example, in 2018, Liu, J. et al. [61] analyzed the applicability of four types of precipitation data, i.e., TRMM 3B42RT, TRMM 3B42V7, CMORPH RAW, and CMORPH CRT, in the simulation of runoff in the Yurungkash River Basin in Xinjiang, and found that TRMM 3B42V7 has a better simulation accuracy. The European Centre for Medium-Range Weather Forecasts' (ECMWF) precipitation data, with a higher spatial accuracy, also achieved higher accuracy in the runoff simulation of the Budhi Gandaki River in Nepal, and the value of $R^2$ was greater than 0.88 [48]. In addition, the calculation method of temperature variables can also be optimized to be more consistent with the natural environmental variability. For example, Li, L.-h et al. [62] used the average effective activity temperature instead of the traditional average daily temperature to exclude the effect of temperature below 0 °C on the average temperature; Muaitar et al. [53] improved the elevation zone division method and the temperature calculation formula, based on the slope and aspect of the watershed. Although the uncertainty of snow accumulation variables stems from the accuracy of remote sensing data, the temporal and spatial resolutions of the source data of this snow accumulation input variable, i.e., MODIS_CGF_SCE,

meet the requirements of snow accumulation variables of the SRM model and better reflect the accumulation and melting process of seasonal snow and permanent snow. When the remote sensing data underestimate the actual snow accumulation area, the recharge of meltwater to runoff is smaller than the actual value, resulting in a lower simulated value than the measured value. Combined with this simulation error analysis, MODIS_CGF_SCE may also have underestimated the snow accumulation area.

In mountainous regions, where snow and meteorological data are sparse, especially in watersheds where no previous work on SRM has been conducted, relevant research under the same region can be of great assistance in improving the efficiency of parameter calibration. What is most critical, is to perform test simulations with sufficient calibration data and, thus, determine the appropriate parameter scheme for the current study basin. For example, instead of the commonly used value of 0.65 for the temperature lapse rate, the results of Ji. X et al. in the Tianshan Mountains, following multi-site data analysis (0.3, 0.67, 0.23), were chosen in this paper, which is more suitable for this basin. However, in terms of the determination of the degree-day factor, the values of other watersheds in the mountainous areas of Xinjiang were basically no higher than 0.36, and the maximum value of the degree-day factor in the Hutubi River Basin was more suitable when it was set to 0.5, after the simulation experiments with regular data of the calibration. Figure 11 shows the results of the 2003–2005 runoff simulation, with a temperature lapse rate of 0.65 and a maximum degree-day factor of 0.36. Table 6 shows the model accuracy coefficients in 2003–2009 with different parameters.

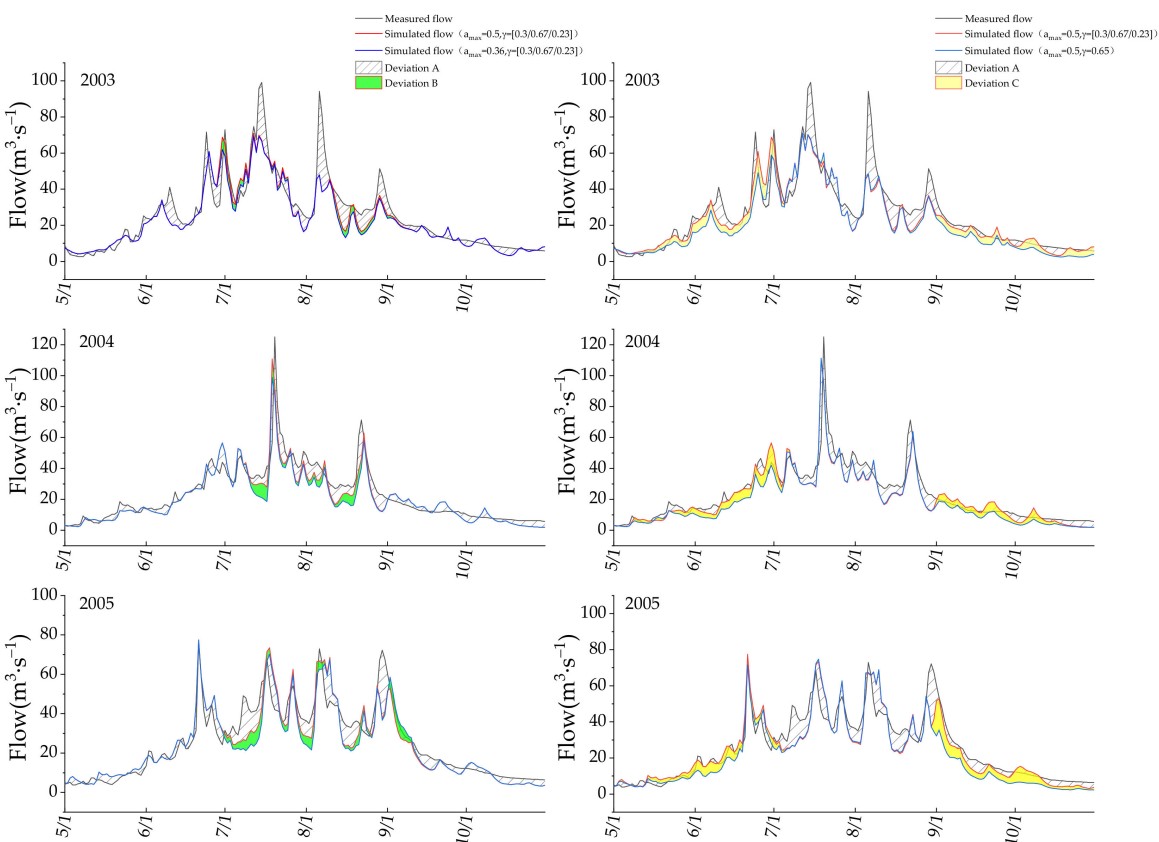

**Figure 11.** Runoff simulation results from 2003 to 2005 with different parameters ($a_{max}$ is the maximum degree-day factor ($cm \cdot °C^{-1} \cdot d^{-1}$), $\gamma$ is the temperature lapse rate (°C /100 m), Deviation A is the variance between the measured flow and the simulated flow for parameter $a_{max}$ of 0.5 and $\gamma$ of 0.3, 0.67, 0.23. Deviation B is the variance between the simulated flow for parameter $a_{max}$ of 0.5 and $\gamma$ of 0.3, 0.67, 0.23 and the simulated flow for parameter $a_{max}$ of 0.36 and $\gamma$ of 0.3, 0.67, 0.23. Deviation C is the variance between the simulated flow for parameter $a_{max}$ of 0.5 and $\gamma$ of 0.3, 0.67, 0.23 and the simulated flow for parameter $a_{max}$ of 0.5 and $\gamma$ of 0.65).

**Table 6.** Model accuracy coefficients in 2003–2009 with different parameters.

| Parameters | Time | $D_V$ | $R^2$ |
|---|---|---|---|
| $a_{max} = 0.5$ $\gamma = 0.3, 0.67, 0.23$ | 2003 | 9.81% | 0.81 |
| | 2004 | 8.55% | 0.82 |
| | 2005 | 4.48% | 0.83 |
| $a_{max} = 0.36$ $\gamma = 0.3, 0.67, 0.23$ | 2003 | 13.1% | 0.79 |
| | 2004 | 13.7% | 0.76 |
| | 2005 | 7.5% | 0.78 |
| $a_{max} = 0.5$ $\gamma = 0.65$ | 2003 | 18.6% | 0.77 |
| | 2004 | 17.9% | 0.80 |
| | 2005 | 14.3% | 0.76 |

The role of glaciers may be underestimated when the degree-day factor takes a value of 0.36, so the simulated values of summer runoff were lower in the overwhelming majority of instances. The effects of temperature lapse rate were more frequently observed in May to June (0.3) and September to October (0.23), which were lower in this simulation than July to August (0.67). When the temperature lapse rate is smaller, the temperature difference between high- and low-elevation areas is narrower, so it is speculated that when high temperature occurs at low-elevation stations, the temperature at high-elevation areas with more snow cover is also higher, snowmelt intensifies and runoff increases. In general, the parameters in the model in this paper refer to the previous research results, and the optimized parameters are determined by the data rate from 2003 to 2005, which is more empirical. Due to the differences in the natural environment in different study areas, the optimized parameter scheme determined in this simulation experiment needs to be further improved. For example, Xie, S. et al. [63] corrected the simulation parameters using the improved segmental optimization algorithm to improve the model simulation accuracy.

This simulation only verified the adaptability of SRM in the Hutubi River Basin but using the model to predict future changes in snowmelt runoff in the context of global warming is more consistent with the expectations of hydrological models. Therefore, while improving the accuracy of the SRM simulations in the Hutubi River Basin, the future change characteristics of this river need to be further investigated.

## 5. Conclusions

The areas below 3200 m elevation in the Hutubi watershed are seasonal snow areas, and above 3200 m are permanent snow areas. The snow cover in the seasonal snow area and permanent snow area has the same trend, and the snow cover of the permanent snow area is usually larger than that of the seasonal snow area. The snowpack in both snow areas starts in October, but the snow increase rate in the permanent snow area is greater than that in the seasonal snow area during the same period. The seasonal snowmelt in the seasonal snow area starts earlier and lasts for a shorter period. The snowmelt time in the seasonal snow area is from March to May, and in the permanent snow area is from May to July, with a minimum snow coverage rate of 0.7%.

The SRM model better simulates the characteristics of runoff changes in the Hutubi River from May to October in 2003–2009. June to August is the high-water period, with a high average daily flow and high frequency of flood peaks. Extreme and continuous precipitation from June to July is one of the influencing factors for the formation of flood peaks. May, September, and October have small and stable flows and are the normal water period. In this period, the frequency of extreme precipitation events is the highest. However, due to the low average temperature and the high probability of extremely low temperature events, the extreme precipitation in May is easily transformed into the snow, and its recharge of runoff continues to play a role in the time series, following the extreme precipitation. During 2003 to 2009, there was an upward trend in the total flow from May to October, with a significant increase in May and a slight decrease in September.

Overall, the simulated value is smaller than the measured value. The Pearson correlation coefficient r between the simulated value and the measured value is 0.83 in the normal water period and 0.77 in the high-water period. The simulation accuracy is higher in the normal water period. The simulation accuracy coefficients $R^2$ are greater than 0.81 in the calibration period, and the mean value of the simulation accuracy coefficient $R^2$ in the validation period is 0.73. The mean value of $D_V$ is 8.85, which is comparable to the total mean accuracy level of the SRM model in other rivers' research results in the Xinjiang region, demonstrating that SRM is suitable for snowmelt runoff simulation in the Hutubi River Basin. To further improve the simulation accuracy, the optimization of input variables and parameters should be further developed, and the prediction of future runoff changes under climate variation is also significant for water resources utilization and management in the Hutubi River Basin.

**Author Contributions:** Conceptualization, X.M., Y.L. and Y.Q.; methodology, X.M. and Y.L.; validation, X.M., Y.L. and Y.Q.; formal analysis, X.M.; investigation, X.M., W.W. and M.Z.; resources, Y.L. and Y.Q.; data curation, X.M., W.W. and M.Z.; writing—original draft preparation, X.M.; writing—review and editing, X.M., Y.L., Y.Q., K.Z. and W.W.; visualization, X.M.; supervision, Y.L. and K.Z.; project administration, Y.L.; funding acquisition, Y.L. All authors have read and agreed to the published version of the manuscript.

**Funding:** This work was jointly supported by the National Key Research and Development Program of China [No. 2019YFC1510505].

**Institutional Review Board Statement:** Not applicable.

**Informed Consent Statement:** Not applicable.

**Data Availability Statement:** The dataset is provided by National Cryosphere Desert Data Center. (http://www.ncdc.ac.cn, accessed on 10 January 2020).

**Acknowledgments:** Special thanks to Yongqiang Liu and his lab members at the College of Geographical Sciences, Xinjiang University, for their work on the paper, and to the National Key Research and Development Program of China (Research and Development of Key Technologies for Snowmelt Flood Warning Decision Support System) for providing research funding support, and to Xiaohua Hao and his team at the Northwest Institute of Eco-Environmental Resources, Chinese Academy of Sciences, for providing research data support. Finally, we extend our hearty gratitude to the anonymous reviewers of this manuscript for their constructive comments and helpful suggestions provided during the preparation.

**Conflicts of Interest:** The authors declare no conflict of interest.

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
