# Peer review of "Adaptability of MODIS Daily Cloud-Free Snow Cover 500 m Dataset over China in Hutubi River Basin Based on Snowmelt Runoff Model"

_sustainability, doi:10.3390/su14074067_

Round 1

Reviewer 1 Report

The article presents an application of SRM (snowmelt runoff model) to the Hutubi River Basin using the MODIS_CGF_SCE dataset. This is a relatively new dataset so evaluating its suitability for snowmelt modelling is definitely an interesting topic. Moreover, the article could provide valuable information for practitioners who are working with SRM or remote sensing data.

There are however a few aspects of the paper that need to be improved. For example, the methods used for calibrating the SRM are not clearly stated. In addition, the section  Uncertainty analysis and optimization could be improved by showing results of the SRM using uncertainty bounds -for example, using different parameters values for the SRM. In addition, the article could also include a comparison between MODIS_CGF_SCE and MOD10A2.

Please also consider the comments included in the attached manuscript pdf file. 

Author Response

  1. The methods used for calibrating the SRM are not stated.

Response:

We sincerely thank you for your valuable comments and questions. We hope that our improvements and interpretations of the manuscript will satisfying you. The parameters of Snowmelt runoff model are strongly related to the natural environmental characteristics of the study area. In this paper, an empirical regression method was used for parameter calibration based on the research of J. Martince et al[1]. The previous research results also have greater reference significance. The methods for determining each parameter are reflected in the paper on Page 8-9, Lines 273 –312.

  1. The section Uncertainty analysis and optimization should show results of the SRM using uncertainty bounds -for example, using different parameters values for the SRM.

Response:

We are very grateful for your valuable comments. We selected two parameters, degree-day factor, and temperature lapse rate, which have a significant impact on the accuracy of the model, for discussion, reflecting the reasonableness of the parameter calibration results in this paper. This content is specifically reflected in the paper on Page 16, Lines 505 – 553.

  1. Discuss the results of simulations using two remote sensing data (MODIS_CGF_SCE/MOD10A2 ).

Response:

Thank you very much for your comments. Your comments are very essential and this is what we are going to do next. In this paper, we mainly focus on the applicability of MODIS_CGF_SCE in Hutubi River Basin and the determination of model parameter schemes, and there is still work for improving the model accuracy. Therefore, we will carry out further work, including remote sensing products comparison.

  1. The comments in the manuscript pdf file.

Response:

Thank you very much for your careful comments in the manuscript, which are valuable for improving the quality of this paper. In response to your comments, we found that there are improper wordings and sentences in the manuscript, and we have corrected them ((line 18, lines 23-24, lines 434-437). Meanwhile, we have added relevant contents, including: the research results of SRM in recent years (lines 82-87, lines 102-122), the model structure graph (lines 186-191, Figure 2), and the graph of the change of hydrological characteristics in July 2007 (lines 456-457, Figure 10).

Thank you again for your valuable comments.

Reviewer 2 Report

I have found the submitted manuscript well-written and well-substantiated. 

Best regards, 

MAA

Author Response

Dear Professor.

We are honored to receive your approval of this article, and your affirmation is the greatest encouragement to our work. We sincerely thank you for taking your valuable time to read our article. All the best to you.

Meng Xiangyao

Reviewer 3 Report

The submitted manuscript discussed the adaptabiligy of "MODIS Daily Cloud-free Snow Cover 500m Dataset over China" for the simulation of snowmelt water runoff in the identified mountainous areas of China. The study is well presented, just some minor comments below.

Minor comments:
- Line 92: Define or descrive "MOD10A2".
- Line 128: Write the not abbreviated form of "DEM" first.
- Equations 1-10: Add reference citations.
- Lines 238: Could not find "L" notations in the presented equations or data.
- Table 3: Provide derivations of these values if possible in the supplementary section.
- Line 449, 458, 470: Remove numbering.

Author Response

Response:

       Thank you very much for your valuable comments, which have helped significantly to improve this article. In this modification, we have added the definition of "MOD10A2" ( lines 97- 98). The full name “Digital Elevation Model” of DEM has been added ( line 159). For equations 1 to 10, we have specified the references in the paper ( lines 273-274 ). The "L" parameter is an empirical value, and the method for determining this value has been described in the paper ( lines 308-310). The method of parameter calibration is given in the paper ( lines 273-312). The serial number has been removed ( line 566, line 575, line 586).

       Thank you again for your contributions to this article.

Reviewer 4 Report

The subject is interesting and important to understand the impact of global warming on the Xinjiang Province. I would advise as below before further consideration for publication 1) Fig. 1, please clearly indicates the location of the study area in China. 2) English are needs carefully improved especially for many grammar problems. 3) The geological characteristics is necessary to assess the climate warming in the time you investigated. Please provide detailed information on local geological information for readers.

Author Response

Response:

We sincerely thank you for your valuable comments and questions. We hope that our improvements and interpretations of the manuscript will satisfying you. The following modifications have been made:

  1. Declared that the study area of this paper is located in China, and added thumbnails of Chinese territory to the figure. ( Figure 1 )
  2. We have revised the misrepresentation in the paper.
  3. We have added a description of geological features in the introduction section of the study area. ( lines 138-145 )

Thank you again for your contributions to this article.

Round 2

Reviewer 1 Report

The authors have addressed all my comments. The section describing the SRM model and its application is now much more detailed. I feel that the manuscript is ready for publication.

This manuscript is a resubmission of an earlier submission. The following is a list of the peer review reports and author responses from that submission.